# High density oilfield wastewater disposal causes deeper, stronger, and more persistent earthquakes

Ryan M. Pollyea [1], Martin C. Chapman[1], Richard S. Jayne [1] & Hao Wu [1]

Oilfield wastewater disposal causes fluid pressure transients that induce earthquakes. Here we show that, in addition to pressure transients related to pumping, there are pressure transients caused by density differences between the wastewater and host rock fluids. In northern Oklahoma, this effect caused earthquakes to migrate downward at ~0.5 km per year during a period of high-rate injections. Following substantial injection rate reductions, the downward earthquake migration rate slowed to ~0.1 km per year. Our model of this scenario shows that the density-driven pressure front migrates downward at comparable rates. This effect may locally increase fluid pressure below injection wells for 10+ years after substantial injection rate reductions. We also show that in north-central Oklahoma the relative proportion of high-magnitude earthquakes increases at 8+ km depth. Thus, our study implies that, following injection rate reductions, the frequency of high-magnitude earthquakes may decay more slowly than the overall earthquake rate.

---

[1] Department of Geosciences, Virginia Polytechnic Institute & State University, 926 West Campus Drive, Blacksburg, VA 24061, USA. Correspondence and requests for materials should be addressed to R.M.P. (email: rpollyea@vt.edu)

The recent boom in unconventional oil and gas production across the midcontinent United States caused a sharp increase in the rate of oilfield wastewater production. This wastewater is discarded by pumping it into deep geologic formations via salt water disposal (SWD) wells[1,2]. The rapid proliferation of SWD operations across the midcontinent United States has been accompanied by collocated and contemporaneous increases in seismic activity[1], particularly in Oklahoma[3] and Kansas[4]. The relationship between SWD operations and seismicity is reasonably explained by the application of effective stress theory to the Mohr-Coulomb failure criterion, which states that effective normal stress acting on a fault decreases in equal proportion to a rise in fluid pressure less any poroelastic relaxation[5]. Thus a sufficient rise of pore fluid pressure in faults optimally aligned to the regional stress field can cause the effective normal stress to drop below the Mohr-Coulomb failure threshold triggering injection-induced earthquakes[6].

Injection-induced earthquakes typically occur 4–8 km below ground surface[7], where direct measurements of pore fluid pressure are rarely, if ever, made before earthquakes are triggered. As a result, physics-based groundwater models have become an effective tool for linking injection-induced fluid pressure changes with earthquake hypocenter locations. One landmark modeling study found that several high-rate SWD wells in southeast Oklahoma City produced a fluid pressure front that accurately matched earthquake hypocenter locations leading up to the Jones earthquake swarm[8]. This history-matching approach was repeated in more recent studies linking SWD operations to earthquake occurrence, e.g., in Milan, Kansas[9], Greeley, Colorado[10], Fairview, Oklahoma[11], Dallas-Fort Worth, Texas[12], and Guthrie, Oklahoma[13]. Based on the success of this history-matching approach, groundwater models are now being incorporated into seismic hazard assessments to simulate fluid pressure decay following SWD volume reductions[14–17]. A common attribute among these and other modeling studies[18–20] is that fluid properties (e.g., density and viscosity) are assumed to be identical between the wastewater and host-rock fluids. However, SWD operations drive pressure transients over km scales into the seismogenic zone, where fluid properties vary substantially due to thermal and geochemical variability. For example, at pressure and temperature conditions representative of ~5 km depth (50 MPa and 100 °C) the density of pure water is 980 kg m$^{-3}$, but for brine composition of 200,000 parts per million NaCl the fluid density is 1120 kg m$^{-3}$ [21] and the viscosity increases ~50%[22].

This study challenges the assumption that fluid properties are of negligible importance to pressure accumulation and decay in the seismogenic zone. We initially consider oilfield wastewater disposal in Alfalfa County, Oklahoma (Fig. 1a, inset), which is located within the Anadarko Shelf geologic province and experienced rapid growth in oil and gas production between 2010 and 2015 as unconventional recovery methods unlocked previously inaccessible resources from the Mississippi Lime formation. Over this same period, Alfalfa County experienced tremendous growth in SWD into the Arbuckle formation and the number of magnitude-2.5 or greater (M2.5+) earthquakes increased from nil in 2010 to over 300 in 2015 (Fig. 1a). Since 2015, both SWD volume and annual earthquake rate have decreased dramatically; however, the mean annual hypocenter depth has been systematically increasing (Fig. 1a). This trend of increasing hypocenter depth years after substantial SWD volume reductions is unexpected because pressure diffusion models show that the rate of pressure accumulation decreases rapidly when injections cease[23]. To explain systematically increasing hypocenter depths in Alfalfa County, we hypothesize

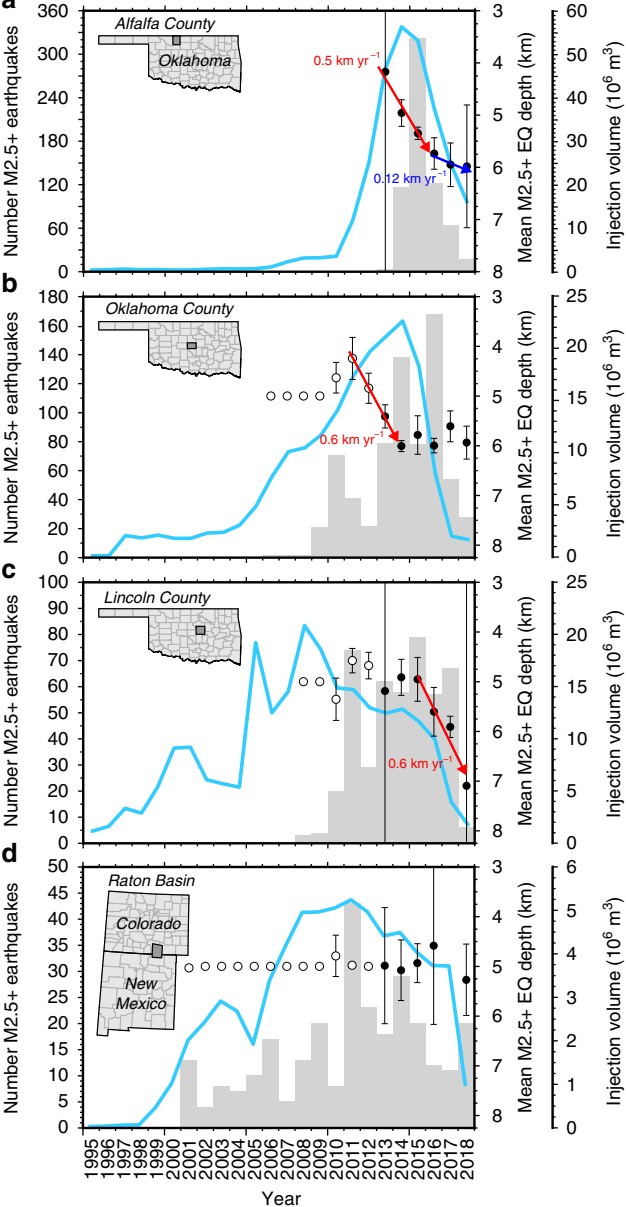

**Fig. 1** Summary earthquake and wastewater data within study areas. Annual M2.5+ earthquake count (gray bars), salt water disposal volume (light blue line), and mean earthquake depth (circles) for **a** Alfalfa, **b** Oklahoma, and **c** Lincoln Counties, which are located in Oklahoma, USA. Panel (**d**) presents the same data for the Raton Basin of Northern New Mexico and Southern Colorado. Mean annual earthquake depth from 2013 to 2018 (solid circles) are weighted by inverse square depth error. Before 2013, mean annual earthquake depths (open circles) are calculated as an arithmetic mean because depth errors were infrequently reported. Error bars correspond with two standard errors. Note that open circles lacking error bars arise because shallow earthquakes were commonly reported to occur at 5 km depth before 2013

that wastewater produced from the Mississippi Lime formation comprises higher total dissolved solids (TDS) concentration (and thus higher density) than fluids in the Precambrian basement (seismogenic zone) and, as a result, the density differential drives advective transport of wastewater into the seismogenic zone, thus increasing fluid pressure enough to trigger earthquakes.

**Table 1 Composition of water produced from Mississippi Lime, Hunton, and Precambrian (basement) formations in select counties of Oklahoma and Kansas**

| Region | State | Formation | Mean TDS[a] (ppm) | $\sigma$ (ppm) | N | Density[b] (kg m$^{-3}$) |
|---|---|---|---|---|---|---|
| Alfalfa Co. | OK | Miss. Lime | 207,000 | 31,000 | 8 | 1123 |
| Grant Co. | OK | Miss. Lime | 235,000 | 30,000 | 54 | 1137 |
| Barber Co. | KS | Miss. Lime | 174,000 | 72,000 | 24 | 1106 |
| Harper Co. | KS | Miss. Lime | 201,000 | 35,000 | 2 | 1120 |
| Sumner Co. | KS | Miss. Lime | 215,000 | 43,000 | 11 | 1127 |
| Lincoln Co. | OK | Hunton | 189,000 | 42,000 | 31 | 1113 |
| Oklahoma Co. | OK | Hunton | 198,000 | 56,000 | 62 | 1118 |
| Central Kansas[c] | KS | Precambrian | 107,000 | 43,000 | 10 | 1068 |
| Raton Basin | CO | Raton Coal[d] | 2000 | 1000 | 800 | 1002 |
| Raton Basin | NM | Precambrian | 70,000 | 18,000 | 5 | 1046 |

Data from USGS National Produced Waters Database[24]
$\sigma$ is one standard deviation
[a]Arithmetic mean
[b]Calculated at 40 °C and 21 MPa[41]
[c]Database records from Rice, Rooks, Rush, Russell, and Barton Counties
[d]Database records for Raton Coal, Raton-Vermejo Coal, Raton Sand-Vermejo Coal, and Raton Sand

## Results

**Composition of wastewater and host-rock fluids**. We analyzed the United States Geological Survey (USGS) National Produced Waters Geochemical Database (NPWGD)[24] and found that oil-field wastewater from the Mississippi Lime formation in northern Oklahoma and southern Kansas comprises higher TDS concentration, and thus higher density, than fluids in the Precambrian basement (Table 1). We then tested the implications of this observation by modeling SWD operations for a typical high-rate (2080 m$^3$ day$^{-1}$) SWD well in Alfalfa County. For this model scenario, the SWD well is open within the upper 200 m of the Arbuckle formation, which occurs between 1900 and 2300 m depth with a permeability of $5 \times 10^{-13}$ m$^2$ (Supplementary Fig. 1a)[25,26]. The Precambrian basement is modeled from 2300 to 10,000 m depth using a dual continua approximation to simulate pressure diffusion and fluid flow through both rock matrix and fractures. In this formulation, 98% of the rock volume is specified as matrix with permeability of $1 \times 10^{-20}$ m$^2$ and the remaining 2% volume comprises the fracture domain. Fracture permeability decreases nonlinearly with depth from $5 \times 10^{-13}$ m$^2$ at the Arbuckle-basement interface (2300 m depth) to $4 \times 10^{-14}$ m$^2$ at 10 km depth (Supplementary Fig. 1b, solid black line). As a result, the volume-weighted effective permeability ranges from $1 \times 10^{-14}$ m$^2$ at the Arbuckle-basement contact to $9 \times 10^{-17}$ m$^2$ at the base of the model domain (Supplementary Fig. 1b, dashed black line). This basement permeability distribution is congruent with estimates for the seismically active crust that suggest bulk permeability is on the order of $\sim 10^{-16}$–$10^{-17}$ m$^2$[27]. The remaining thermal and hydraulic model parameters are presented in Supplementary Table 1. For this model, the TDS concentrations of injected wastewater and host-rock fluids were based on mean USGS NPWGD values for water produced in the Mississippi Lime formation in Alfalfa County, Oklahoma and Precambrian basement in central Kansas, respectively (Table 1). Our model solves the conservation equations for energy and mass transport[28], and takes into account the regional geothermal gradient, temperature- and pressure-dependent fluid properties, and fluid mixing by advection and diffusion. We simulated a 10 year injection period followed by 40 years of pressure recovery. We then compare this non-isothermal variable density model with an identically parameterized model that neglects thermal and compositional variability between the wastewater and host-rock fluids. To account for uncertainty in the basement fracture permeability, we tested two additional model scenarios comprising lower fracture permeability in the basement (Supplementary Fig. 1b).

**Density-driven pressure accumulation is persistent**. In the vicinity of the injection well, our models show that pressure accumulation after one year of injection is indistinguishable between the variable- and constant-density models (Fig. 2a, d). This initial rise in fluid pressure is caused by the addition of fluid mass to the system, which increases the dynamic load and drives pressure diffusion through the interconnected fracture network. After five years of SWD operations, the effects of wastewater injection are apparent as a slug of high-density brine migrating vertically downward to ~4 km depth (Fig. 2b). Since fluid pressure increases linearly with fluid density, this advective transport of high-density wastewater locally increases fluid pressure as wastewater displaces lower density basement fluids. The development of this density-driven pressure front is further enhanced by the natural geothermal gradient because the sinking wastewater plume passes through systematically warmer (and lower density) host-rock fluids. After 10 years of injection, the density driven pressure front exceeds 70 kPa at ~6 km depth, while also expanding ~2 km laterally to increase fluid pressure above 20 kPa (Fig. 2c). In contrast, the constant-density model only accounts for pressure diffusion and the corresponding pressure increase after 5 and 10 years of injection is just 10 kPa at 5 and 6 km depth, respectively (Fig. 2e, f).

Figure 3 presents timeseries results of fluid pressure increase above background conditions ($\Delta P_f$ in kPa) for the variable- and constant-density models at monitoring points located directly below the injection well. These results show that $\Delta P_f$ below the injection well is generally independent of fluid density during the first two years of injection operations. After this time, the arrival of high-density wastewater appears at 4 km depth as a sharp upward change in the $\Delta P_f$ curve (Fig. 3). This phenomenon occurs at 5 and 6 km depth on a time lag of ~2 years each. As a result, the rate of advective wastewater migration through the seismogenic zone is ~0.5 km yr$^{-1}$. Furthermore, this advective fluid migration increases the pressure accumulation rate to ~20 kPa yr$^{-1}$ as the wastewater plume passes through the seismogenic zone. In contrast, the $\Delta P_f$ curve for the constant-density model shows that fluid pressure rapidly increases for two years, and then plateaus to a negligible pressure accumulation rate (Fig. 3, dashed lines).

When SWD operations cease after 10 years of injection, our timeseries results show that maximum fluid pressure accumulation (~80 kPa) occurs at 6 km depth and 3 years after injection stops (Fig. 3). These results also reveal that high TDS wastewater continues downward migration through the

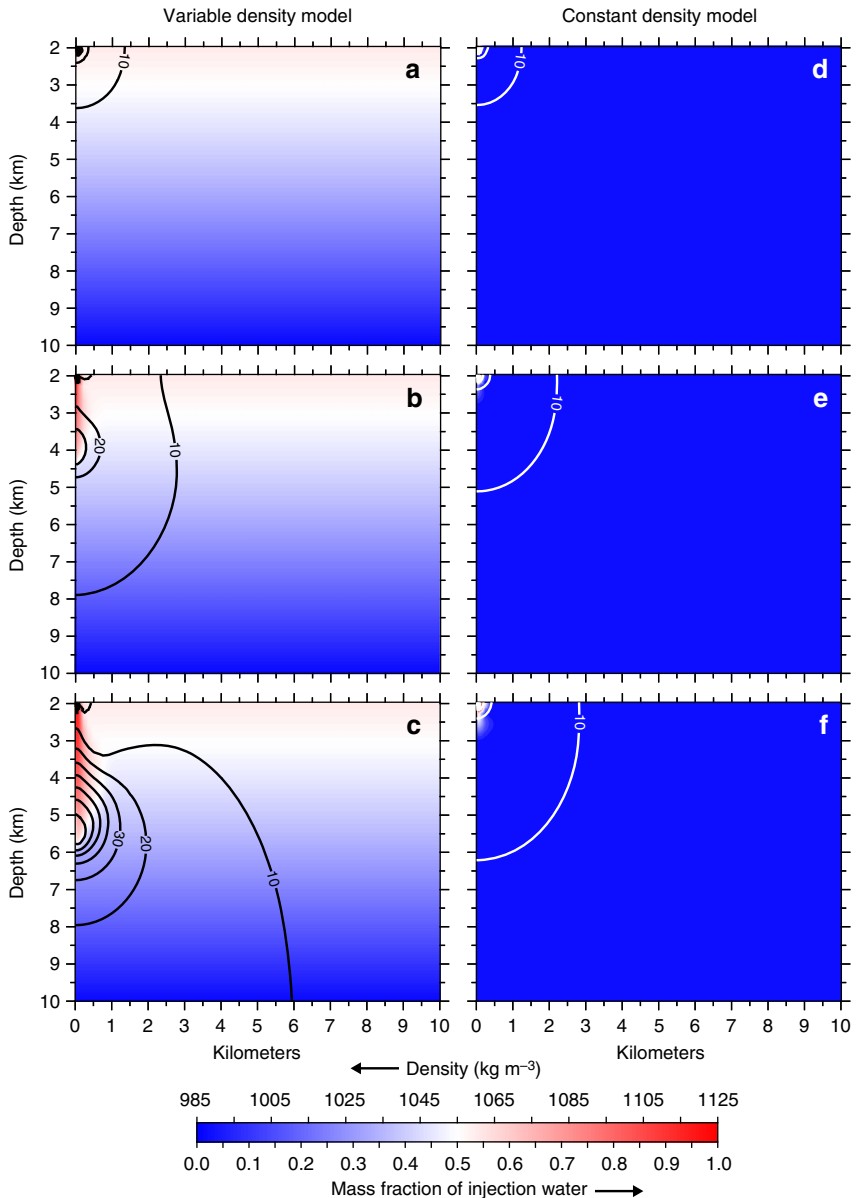

**Fig. 2** Models of variable- and constant-density salt water disposal. Results for a single salt water disposal well operating for 10 years at 2,080 m³ day⁻¹ (13,000 bbl day⁻¹). Left column shows variable density results after **a** 1 year, **b** 5 years, and **c** 10 years of injection. Black contour lines denote fluid pressure change above initial conditions in 10 kPa increments. Background shading for the variable density models is fluid density, which decreases with depth due to increasing temperature. Right column shows constant-density results after **d** 1 year, **e** 5 years, and **f** 10 years of injection. White contour lines denote fluid pressure change above initial conditions in 10 kPa increments, and background shading is the mass fraction of injection water

seismogenic zone for an additional decade causing increasing fluid pressure at sequentially greater depths (Fig. 4a–c). Because there is no additional wastewater injection to increase the dynamic load, this post-injection pressure accumulation is due solely to the advective transport of the high-density wastewater. During this post-injection period, the rate of advective waste-water migration decreases to ~0.18 km yr⁻¹ (Fig. 3). This is due to a combination of lower fracture permeability at depth, lack of additional fluid mass from injection, and mixing between wastewater and host-rock fluids. Because this result has significant implications for long term injection-induced earthquake hazard, we tested two additional basement perme-ability models (Supplementary Fig. 1b). We found that the fluid pressure recovery rate and depth of maximum fluid pressure are modulated by permeability and that post-injection fluid pressure in the seismogenic zone continued

increasing for up to 15 years in both supplemental cases (Supplementary Fig. 2).

Pressure recovery rates predicted by these variable density models are substantially different than pressure recovery in the constant-density models, the latter of which recovers rapidly to pre-injection conditions (Supplementary Fig. 2). Our complete set of results (Figs 2–4, Supplementary Figs. 2–7) reveal that high-density wastewater can become effectively trapped within the seismogenic zone thus maintaining elevated fluid pressure over 10- to 15-year timescales. In aggregate, our results show that the density-driven pressure accumulation and decay process is robust to a wide range of permeability scenarios when there is a large density contrast between SWD and basement fluids.

**Density-driven pressure transients cause deeper earthquakes**. In 2015, there were ~85 SWD wells operating in the Arbuckle and

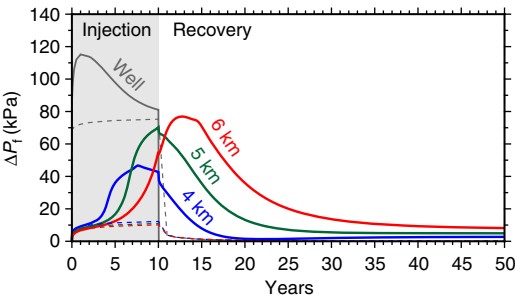

**Fig. 3** Time series of fluid pressure above initial conditions ($\Delta P_f$). Simulation data are recorded at monitoring points located within the injection well (black) and directly below the well at 4 km (blue), 5 km (green), and 6 km (red) depth. Solid and dashed lines are $\Delta P_f$ curves for the variable density and constant-density models, respectively. For a given depth, the difference between solid and dashed lines are due to the advective transport of high-density wastewater

Simpson formations within Alfalfa County, Oklahoma with an average injection rate of 1399 m³ day⁻¹ (8746 bbl day⁻¹)[3]. These wells were separated by an average nearest-neighbor well spacing of ~1.5 km (min = 0.05 km, max = 8.9 km, $\sigma$ = 1.7 km). By 2017, the average SWD rate had fallen to 853 m³ day⁻¹ (5333 bbl day⁻¹), but the average distance between SWD wells remained less than 2 km[3]. Our variable density model shows that the density-driven pressure front expands ~2 km laterally within several years (Fig. 2b, c). As a consequence, the combination of high TDS wastewater and numerous closely spaced injection wells in Alfalfa County suggests that an areally extensive slug of high-density wastewater is driving fluid pressure systematically deeper within the seismogenic zone. This explains why mean annual hypocenter depths have been increasing each year since the onset of seismicity (Fig. 1a). This deepening trend occurred at a rate of ~0.5 km yr⁻¹ between 2013 and 2015, and began after three years of rapidly increasing SWD volume (Fig. 1a). This timing is consistent with our model results which show that pressure accumulation from advective brine transport reaches 4 km depth after ~2–3 years of injection and brine passes through the seismogenic zone at a rate of ~0.5 km yr⁻¹ during injections (Fig. 3). Following several years of substantial reductions in wastewater injection volume, the number of earthquakes decreased dramatically while mean annual hypocenter depth continued increasing at a slower rate of ~0.12 km yr⁻¹ (Fig. 1a). This response to reduced wastewater injection volume is also congruent with our model scenario, which shows that the rate of brine migration through the seismogenic zone decreases to ~0.18 km yr⁻¹ during the post-injection phase (Fig. 3).

East of the Nemaha fault zone in central Oklahoma, mean annual hypocenter depths in Lincoln and Oklahoma Counties also increased at rates comparable to our simulation results (Fig. 1b, c). In this region, SWD fluids originate largely from the Hunton dewatering play and are characterized by mean TDS concentrations of 189,000 ppm and 198,000 ppm in Lincoln and Oklahoma Counties, respectively (Table 1). We also tested the alternative case in which wastewater is characterized by lower TDS concentration, and thus lower density, than fluids in the seismogenic zone. For this test, we considered the Raton Basin of southern Colorado and northern New Mexico (Fig. 1d), where wastewater injections associated with coal-bed methane production have been implicated in regional earthquake occurrence since at least 2008[29]. Within the Raton Basin, the USGS NPWGD shows that wastewater produced along with coal-bed methane is characterized by a mean TDS concentration of just 2000 ppm[24], so it has a lower density than basement fluids (Table 1). As a

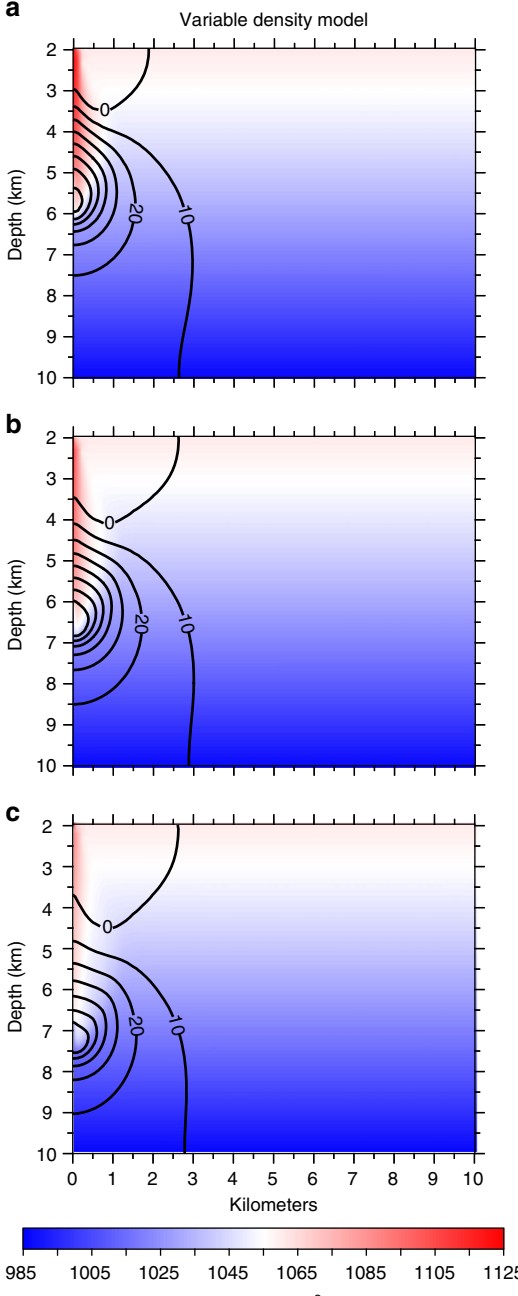

**Fig. 4** Post-injection fluid pressure recovery. Simulation results showing post-injection fluid pressure patterns for the variable density model following ten years of wastewater disposal at 2080 m³ day⁻¹ (13,000 bbl day⁻¹). Remaining fluid pressure above background ($\Delta P_f$) is shown after **a** 1 year, **b** 5 years, and **c** 10 years of post-injection fluid pressure recovery. Contour lines denote $\Delta P_f$ in 10 kPa increments. The corresponding constant-density model recovers to <10 kPa within 1 year post-injection, and the results are presented in Supplementary Materials

consequence, the difference in fluid potential energy is unfavorable for downward, density-driven pressure accumulation, and the mean annual hypocenter depths do not exhibit a systematically increasing trend (Fig. 1d).

**Deeper earthquakes may be larger.** To assess the potential for advective brine transport to affect earthquake magnitude distributions in northern Oklahoma and southern Kansas (Fig. 5a),

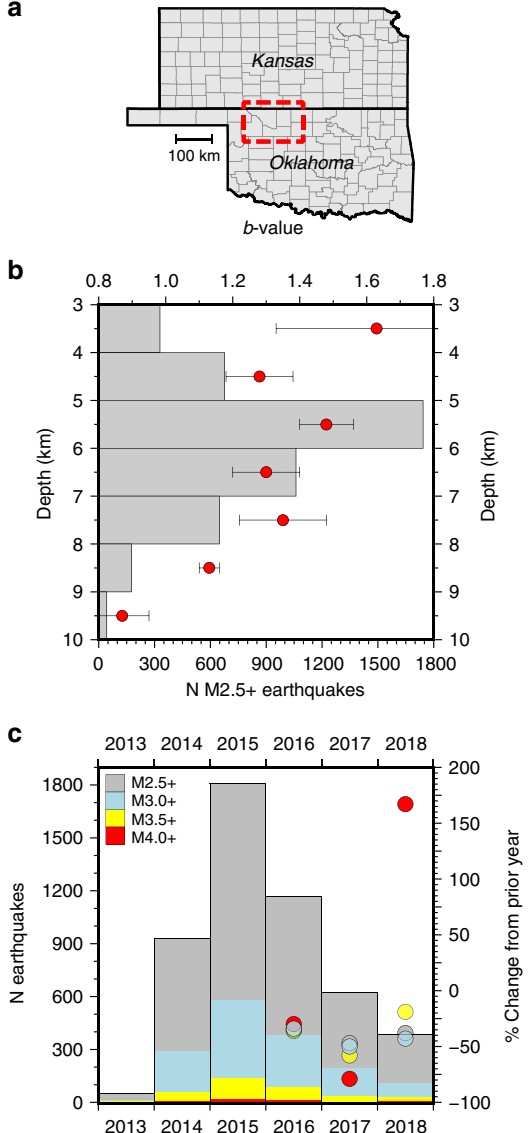

**Fig. 5** Earthquake depth-magnitude analysis between 1 Jan 2013 and 31 Dec 2018. **a** Areal extent of the earthquake catalog within northern Oklahoma and southern Kansas. **b** Earthquake distribution in 1 km depth intervals (gray bars) with the corresponding Gutenberg-Richter *b*-value shown as red circles. The *b*-value systematically decreases beyond 8 km depth indicating that the relative proportion of high-magnitude earthquakes is larger in the 8–10 km depth interval. Error bars correspond with two standard errors of the regression slope for each depth interval. **c** Annual distribution of M2.5+ (gray), M3.0+ (blue), M3.5+ (yellow), and M4.0+ (red) earthquakes, as well as the percent change from the prior year (circles) beginning when the overall earthquake started to decline

we calculated the Gutenberg-Richter *b*-value at 1 km depth intervals for the time period 2013–2018 (Supplementary Fig. 8). The *b*-value characterizes the shape of the cumulative frequency-magnitude distribution, for which the relative proportion of high-magnitude earthquakes increases as the *b*-value decreases[30]. Our results show that from 2013 to 2018 the *b*-value oscillates between 1.63 and 1.28 at depths less than 8 km, and then systematically decreases to 0.87 between 8 and 10 km depth (Fig. 5b). These results are in general agreement with Mori and Abercrombie[31], who found that the *b*-value for earthquakes in northern California systematically decreases from 1.28 between 0 and 3 km

depth to 0.87 between 9 and 12 km depth. The implication of lower *b*-values at 8+ km depths is that the proportion of high-magnitude earthquakes increases relative to the earthquake distribution at shallow depths.

Our model shows that density-driven pressure transients may locally increase fluid pressure at 8+ km depth for over a decade after SWD operations cease (Fig. 4c). As a consequence, earthquakes triggered by advective brine transport after substantial SWD rate reductions are likely to have a larger relative proportion of high-magnitude earthquakes despite a lower overall earthquake rate. In 2018, this phenomenon was manifest in northern Oklahoma and southern Kansas as a dramatic year-over-year increase in the number of M4+ earthquakes despite lower year-over-year occurrence rates for smaller magnitude earthquakes (Fig. 5c). Thus, density-driven pressure transients may explain why Oklahoma experienced three M5+ earthquakes in 2016 despite the implementation of earthquake mitigation measures mandating widespread SWD volume reductions[32]. Moreover, the persistence of density-driven fluid pressure transients may help to explain why the USGS one-year seismic hazard forecast found that the probability for damaging ground motion in Oklahoma and Kansas increased from 2017 to 2018 despite the broad trend of declining overall earthquake rates that began in 2016[33].

## Discussion

Injection-induced earthquakes are caused by fluid pressure transients that decrease effective normal stress on optimally-oriented faults. As a consequence, managing the hazard posed by injection-induced earthquakes requires fundamental knowledge about the hydrogeological processes governing both fluid pressure accumulation during SWD operations and fluid pressure recovery following injection rate reductions. To date, much of the regulatory response to injection-induced earthquake hazard mitigation requires SWD volume reductions for wells in close proximity to earthquake swarms[3]. This earthquake mitigation strategy is congruent with the classical root-time scaling law for pressure diffusion[34], and it is supported by numerous modeling studies showing that fluid pressure responds rapidly to both SWD injections[8–13,18–20] and SWD injection rate reductions[14–17]. However, these former studies each assume that density differences are negligible between wastewater and host-rock fluids. This assumption further implies that pressure diffusion is the only process capable of inducing pressure transients, and thus triggering earthquakes, during SWD operations.

Here we show that density-driven fluid flow also affects fluid pressure accumulation and recovery when oilfield wastewater has a higher TDS concentration than the basement fluids, as is the case throughout Oklahoma and southern Kansas. This causes the higher density wastewater to travel vertically downward into the seismogenic zone, which increases fluid pressure in the basement rocks as high-density wastewater displaces lower density host-rock fluids. Our model of this process shows that density-driven fluid transport can delay pressure recovery and even lead to increasing fluid pressure long after significant injection rate reductions (Fig. 3, Supplementary Fig. 2). This means that in regions with high TDS wastewater, e.g., in Oklahoma and Kansas, earthquake rates may either decline more slowly than current models predict[14–17] or remain above historic averages for years after volume reductions. We found evidence for these advective pressure transients in the systematically increasing earthquake hypocenter depths in Oklahoma (Fig. 1a–c), where oilfield wastewater is characterized by TDS concentration between 174,000 and 235,000 ppm (Table 1). Moreover, this deepening hypocenter trend occurred in Alfalfa and Lincoln

Counties during a period of rapid injection rate reductions, when the effects of pressure diffusion were rapidly decaying. In contrast, we showed that earthquake hypocenter depths do not exhibit a depth trend in the Raton Basin, where wastewater has a lower density than basement fluids (Fig. 1d).

In the context of hazard mitigation strategies, our results suggest that the local effects of advective brine transport into the seismogenic zone delays fluid pressure recovery over 10+ year timescales. This means that earthquake mitigation strategies, predictive hazard models, and risk assessment procedures should consider the effects of fluid density contrast in addition to pressure diffusion induced by dynamic loading. Recent studies coupling the effects of fluid pressure propagation with earthquake hazard[14,17] and occurrence[15,16] in Oklahoma and Kansas make the reasonable assumption of uniform fluid composition during the injection phase of SWD operations when dynamic loading is the primary process governing fluid pressure accumulation. Our study indicates that such models can be enhanced to consider how advective brine transport governs pressure recovery after widespread SWD rate reductions. This enhancement is particularly important because we also found that the relative proportion of high-magnitude earthquakes increases at 8+ km depths in northern Oklahoma and southern Kansas (Fig. 5). Because fluid pressure continues increasing at these depths for over a decade after significant SWD rate reductions (Fig. 4c), our study implies that even though earthquake frequency may decline after reduction of SWD injection rates, the sinking wastewater may induce larger earthquakes. Put differently, mandated SWD rate reductions have effectively decreased the number of injection-induced earthquakes in Oklahoma and Kansas, but the occurrence rate of high-magnitude earthquakes is decreasing more slowly than the overall earthquake rate (Fig. 5c) because density-driven pressure transients remain in the environment for much longer time periods than those governed by pressure diffusion.

In closing, we note that recent global estimates of unconventional fossil fuel resources exceed 440 billion tons oil and 227 trillion cubic meters gas from 363 petroleum basins worldwide[35]. As these resources are developed, results from this study suggest that the density contrast between produced waters and basement fluids is a fundamental component of the risk profile for injection-induced earthquakes during oilfield wastewater disposal in deep geologic formations. We hope this study motivates further research into the relationship between fluid properties and injection-induced seismicity.

## Methods

**Data sources**. Earthquake data were acquired by internet download from the United States Geological Survey ComCat earthquake catalog[36] on 21 February 2019. These data were acquired in four separate downloads for M2.5+ earthquakes occurring between 1 January 1995 and 31 December 2018. For Alfalfa County, Oklahoma, the search criteria comprise geographic coordinates 36.4623° to 37.0000° and −98.5426° to −98.1038°. For Lincoln County, Oklahoma, the search criteria comprise geographic coordinates 35.4269° to 35.9397° and −97.1388° to −96.6199°. For Oklahoma County, Oklahoma, the search criteria comprise geographic coordinates 35.3773° to 35.7250° and −97.6739° to −97.1417°. For the Raton Basin, the search criteria comprise geographic coordinates 36.7571° to 37.6310° and −104.9960° to −104.1960°. Data for earthquake frequency-magnitude analysis for north-central Oklahoma and southern Kansas was downloaded from the USGS ComCat earthquake catalog[36] on 21 February 2019 using the geographic coordinates 36.125° to 37.235° and −97.800° to −99.602°, M2.5+ magnitude threshold, and date range 1 January 2013 to 31 December 2018.

SWD data for the Raton Basin in northern New Mexico were acquired by internet download from the New Mexico Oil Conservation Division[37] on 7 June 2018. The NMOCD permits county-specific queries, and we selected Colfax County for all available years, 2006–2018. These data are current through 1 March 2018. Wastewater injection data for the Raton Basin in southern Colorado were acquired by internet download from the Colorado Oil and Gas Conservation Commission[38] on 7 June 2018. For this query, we selected a facility inquiry for Las Animas County with facility type equal to "UIC Disposal" and "UIC Simultaneous Disposal". A total of 25 wastewater injection wells were returned by the database, of

which 22 comprised non-zero records, and annual total volume was compiled and merged for each well. Saltwater disposal data for Oklahoma between 2017 through 2018 were acquired by internet download from the Oklahoma Corporation Commission (OCC) Oil and Gas datafiles[39]. These data comprise weekly reports of daily injection volume, and county-level aggregation was performed by coordinate-based filtering using the same search criteria as described above for the earthquake data. Saltwater disposal data for Oklahoma between 2011 through 2016 were acquired by internet download from the OCC Oil and Gas datafiles[40], and volume data between 1995 and 2010 were provided by email request from OCC. These latter datasets permit county-level search.

The USGS National Produced Waters Database[24] was acquired by internet download on 13 June 2018. This database aggregates geochemical data by state, county, and geologic formation. To identify sample results relevant for this study, we implemented the following database search queries:

- State: "Kansas"

County: "Barber", "Harper", "Sumner"

Formation: "Mississippi", "Mississippi Chat", "Mississippian"

- State: "Oklahoma"

County: "Alfalfa", "Alfalfa and Grant"

Formation: "Mississippian", "Mississippi", "Mississippian Meramec"

- State: "Colorado"

Formation: "Raton Coal", "Raton-Vermejo Coal", "Raton Sand - Vermejo Coal", "Raton Sand"

- Formation: "Precambrian"

State: "Kansas", "New Mexico"

For each query, the record labeled "TDSUSGS" was used for analysis.

**Oilfield wastewater and basement fluid composition**. The USGS NPWD indicates that brine produced in Mississippi Lime in Alfalfa County is characterized by mean TDS concentration of ~207,000 ppm ($\sigma$ = 31,000 ppm, $N$ = 8)[24]. Assuming that this value is representative of modern oilfield wastewater allows us to estimate an average brine density for SWD fluid in Alfalfa County of 1123 kg m$^{-3}$ ± 15 kg m$^{-3}$ [41]. This estimate assumes that the TDS are primarily NaCl and SWD occurs at the temperature (40 °C) and pressure (21 MPa) conditions typical of the disposal reservoir (Arbuckle formation). This latter assumption is conservative because produced waters are likely to cool during withdrawal and separation, thus increasing density. Since the USGS NPWD includes only eight records with TDS data for Mississippi Lime formation in Alfalfa County, we tested the robustness of our estimated TDS value by calculating mean TDS concentration and corresponding density for produced water from the Mississippi Lime within four additional counties on the Anadarko Shelf. We found a range of 174,000–235,000 ppm, which corresponds with a density range of 1106–1137 kg m$^{-3}$ (Table 1). Currently, TDS values for Precambrian basement fluids in Oklahoma are unavailable; however, the USGS NPWD includes 10 records for Precambrian basement fluids in central Kansas. The mean TDS concentration for these records is 107,000 ppm ($\sigma$ = 48,000 ppm)[24], which corresponds with mean fluid density of 1068 kg m$^{-3}$ ± 30 kg m$^{-3}$ at 21 MPa and 40 °C[41].

In the Raton Basin, USA, wastewater injections associated with coal-bed methane production have been implicated in regional earthquake occurrence since at least 2008[29]. In this region, water produced with coal-bed methane in southern Colorado is characterized by a mean TDS concentration of 2000 ppm ($\sigma$ = 1200 ppm, $N$ = 800)[24]. This fluid composition suggests that wastewater TDS in the Raton Basin is lower than the TDS concentration of basement fluids, thus implying lower fluid density (Table 1).

**Conceptual model**. The conceptual model for this study represents the Arbuckle formation in northern Oklahoma, which occurs from 1900–2300 m depth[42] and overlies the Precambrian basement that we model to a depth of 10,000 m (Supplementary Fig. 1a). The model domain is 200 km × 200 km laterally; however, we invoke fourfold symmetry to reduce the simulation grid to a lateral extent of 100 km in each direction (Supplementary Fig. 1a). We modeled a single, SWD well that operates at 2080 m$^3$ day$^{-1}$ (13,000 bbl day$^{-1}$), which is below the maximum allowable rate (15,000 bbl day$^{-1}$) for wells on the Anadarko Shelf in Oklahoma[43]. We further simulated two fluid composition scenarios. The first scenario accounts for nonisothermal variable density groundwater flow, and both SWD and basement fluid compositions were specified in accordance with entries in Table 1 for Alfalfa

County, Oklahoma and central Kansas, respectively. The second fluid composition scenario is isothermal with uniform fluid properties. For each simulation, SWD injections occur in the upper 200 m of the Arbuckle formation for 10 years at constant rate, and the simulations continue for an additional 40 years to monitor fluid pressure recovery.

**Model grid discretization.** The 100 km × 100 km × 8.1 km volume is modeled as a 3-D unstructured grid comprising 1,278,613 total grid cells with local grid refinement near the SWD well. The Precambrian basement is discretized as multiple interacting continua (MINC) to separately account for fracture and matrix flow[44]. The conceptual model for MINC discretization is based on an assemblage of matrix blocks embedded in a fracture network, and we invoke this model by further discretizing each basement grid cell into both fracture and matrix continua[44]. In this formulation, the fracture continuum is hydraulically connected to the overall fracture network, while the matrix continuum only maintains hydraulic connectivity with its local fracture network. As a result, MINC discretization permits only fracture-fracture and fracture-matrix flow between grid cells. We parameterize the MINC formulation by assuming parallel-plate fractures in three-coordinate directions and assigning 2% of the total grid cell volume to the fracture continuum, while the matrix continuum comprises the remaining 98% volume percent.

**Model parameters.** To account for uncertainty in basement fracture permeability, we consider three fracture permeability scenarios that each decay with depth according to the Manning and Ingebritsen[45] relation: $k(z) = k_0(z/z_0)^{-3.2}$. For our models, $k_0$ and $z_0$ are permeability and depth at the Arbuckle-basement interface (2300 m depth), respectively. The primary permeability scenario discussed in main text corresponds with $k_0$ of $5 \times 10^{-13}$ m$^2$. This results in bulk effective permeability ranging from $1 \times 10^{-14}$ m$^2$ to $9 \times 10^{-17}$ m$^2$ across the thickness of the Precambrian basement. These effective permeability values are congruent with basement permeability values reported in the literature for northern and central Oklahoma[8,11]. We additionally test two permeability scenarios characterized by lower fracture permeability than the primary scenario. These permeability scenarios are specified by $k_0$ of $1 \times 10^{-13}$ m$^2$ and $5 \times 10^{-14}$ m$^2$, and we label them permeability scenarios B and C, respectively. A comparison of all three permeability scenarios is shown in Supplementary Fig. 1b. The remaining hydraulic and thermal properties for each rock type are presented in Supplementary Table 1.

**Model initial and boundary conditions.** Initial conditions for all model scenarios comprise a hydrostatic gradient, which is calculated separately for the variable and constant-density models. For the variable density models, the initial temperature distribution is calculated on the basis of a 40 mW/m$^2$ heat flux across the bottom of the domain[46], which results in a geothermal gradient of 18 °C/km. The initial pressure distribution for the constant-density model is calculated on the basis of 52 °C uniform temperature. Boundary conditions for all simulations comprise constant pressure (and temperature for the variable density model) in the far-field to prevent non-physical pressure feedbacks from the lateral boundaries, no-flow boundaries across the top and bottom of the domain, and a 40 mW/m$^2$ heat basal heat flux for the variable density simulations. We also invoke four-fold symmetry in the model domain, thus no-flow boundaries are specified in the $xz$- and $yz$-planes through the origin for all simulations.

**Model governing equations.** The code selection for this study is TOUGH3[28] compiled with equation of state modules EOS7 and EOS1 for simulating non-isothermal mixtures of brine and pure water, respectively, as well as mixing by chemical diffusion. The TOUGH3 simulator solves the conservation equations for mass and energy flow in porous geologic media. The complete solution scheme is presented in the TOUGH3 User's Guide[47], and we summarize the governing equations in the context of fully saturated flow here. The general form of mass and energy conservation equations is written as:

$$\frac{d}{dt}\int_{V_n} M^\kappa dV_n = \int_{\Gamma_n} \mathbf{F}^\kappa \cdot \mathbf{n} d\Gamma_n + \int_{V_n} q^\kappa dV_n \quad (1)$$

In this formulation, the left side of equation 1 is the accumulation term, where $M$ represents a mass (or energy) component $\kappa$ which for this study are water, brine and/or energy (in which case $\kappa$ is specific inner energy). As a result, the time-change of mass (or energy) within closed volume $V_n$ is equivalent to the sum of the integral component flux ($\mathbf{F}^\kappa$) normal to the volume-bounding surface ($\Gamma_n$), as well as any sources or sinks ($q^\kappa$) of component $\kappa$ within $V_n$.

The mass accumulation term in equation 1 is generalized as:

$$M^\kappa = \varphi \sum S_\beta \rho_\beta X_\beta^\kappa \quad (2)$$

where $\varphi$ is porosity, $S_\beta$ is the saturation of phase $\beta$ (only aqueous phase is considered in this study), $\rho_\beta$ is density of phase $\beta$, $X_\beta^\kappa$ is mass fraction of mass component $\kappa$ in phase $\beta$. In Eq. 2, $M^\kappa$ is summed over all fluid phases occupying pore space in $V_n$; however, for this study, we are only considering fully saturated flow. For the variable density models that account for nonisothermal groundwater

flow, the heat accumulation term, is given by:

$$M^\kappa = (1-\varphi)\rho_R C_R T + \varphi \sum S_\beta \rho_\beta u_\beta \quad (3)$$

where $\rho_R$ is rock density, $C_R$ is rock specific heat, T is temperature, and $u_\beta$ is enthalpy of phase $\beta$. In TOUGH3, the advective flux ($\mathbf{F}^\kappa|_{adv}$) for each mass component $\kappa$ is given as the sum of all phase fluxes, $\mathbf{F}^\kappa|_{adv} = \sum X_\beta^\kappa \mathbf{F}_\beta^\kappa$, where $\mathbf{F}_\beta$ is presented here in terms of Darcy's Law for fully saturated porous media:

$$\mathbf{F}_\beta = -\frac{k\rho_\beta}{\mu_\beta}(\nabla P_\beta - \rho_\beta \mathbf{g}) \quad (4)$$

where $k$ is intrinsic permeability, $\mu_\beta$ is dynamic viscosity of phase $\beta$, $P_\beta$ is fluid pressure of phase $\beta$, and $\mathbf{g}$ is the vector of gravitational acceleration. Diffusive mass transport ($\mathbf{f}^\kappa$) is modeled as,

$$\mathbf{f}^\kappa = -\varphi \tau_0 \tau_\beta \rho_\beta D_\beta \nabla X_\beta^\kappa \quad (5)$$

where $\tau_0 \tau_\beta$ is the tortuosity coefficient (not considered in our models) and $D_\beta^\kappa$ is the diffusion coefficient for mass component $\kappa$ in phase $\beta$. Our models consider SWD wells as source terms in the relevant grid cells, for which a constant mass rate of either pure water or brine is specified. To convert from volume rate ($Q$) to mass rate ($\dot{m}$), we use the standard conversion, $\dot{m} = Q\rho$, where $\rho$ is the injection fluid density at reservoir temperature and pressure.

In TOUGH3, the governing equations are solved by the integral finite difference method for space discretization, while time discretization is fully implicit, first-order backward finite difference. This results in a coupled, nonlinear set of equations that are solved simultaneously by Newton-Raphson iteration. We utilize automatic time step control for the Newton-Raphson iterations by doubling the time step when convergence is achieved within four iterations and halving the time step when convergence requires eight iterations. For nonisothermal simulation, the temperature dependence on properties of pure water are calculated internally from the steam equations.

**Constant-density model.** The constant-density model scenarios are calculated with TOUGH3 using the equation of state module, EOS1[47]. For these simulations, we invoke the "two-waters" function of EOS1 in order to track the fate SWD injection. In the two-waters formulation, individual mass balances are solved for each water component, while maintaining identical thermophysical water properties within each cell. We invoke this function to specify initial reservoir fluids as "water 1" and SWD fluids as "water 2", which provides a mechanism for tracing SWD fluids through the system on the basis of the mass fraction of water 2. As a result, the spatial distribution of SWD fluids is visualized by contouring the mass fraction of injected water.

**Variable density model.** To solve the variable density scenarios in which SWD comprises higher TDS concentration than basement fluids, we implement the TOUGH3 equation of state module for aqueous mixtures of pure water and brine, EOS7[47]. In this formulation, aqueous phase salinity is accounted for on the basis of a brine mass fraction, $X_b$, and density and viscosity are interpolated between end-members comprising pure water and brine. Although the code makes allowances for unsaturated conditions, we consider only fully saturated flow in this study, and, as a result, our models obviate problems that may be encountered during phase change (e.g., salting out effects). The fundamental assumption in EOS7 is that fluid volume is conserved during mixing of water and brine[48]. As a result, the density of the water-brine mixture ($\rho_m$) for variable brine saturation ($X_b$) can be approximated as,

$$\frac{1}{\rho_m} = \frac{1-X_b}{\rho_w} + \frac{X_b}{\rho_b} \quad (6)$$

where $\rho_w$ is the density of pure water and $\rho_b$ is the density of a reference brine when $X_b$ is one. For our study, the reference brine density is 1123 kg/m$^3$. The approximation for density of the brine-water mixture (Eq. 6) further assumes the compressibility of brine to be the same as for pure water. To account for the effects of pressure, temperature, and salinity on the viscosity of the brine-water mixture ($\mu_m$), the polynomial correction by Herbert[48] is invoked as:

$$\mu_m(P, T, X) = \mu_w(P, T)\left[1 + 0.4819X_b - 0.2774X_b^2 + 0.7814X_b^3\right]. \quad (7)$$

where $\mu_w$ is the viscosity of pure water, for which temperature and pressure dependence is accounted for by internally referencing the equation of state for water.

**Earthquake depth calculations.** Mean annual earthquake depth is calculated for Alfalfa, Oklahoma, and Lincoln Counties, Oklahoma, as well as the Raton Basin of southern Colorado and northern Kansas. We perform this calculation for earthquakes between 3 and 10 km depth using two methodologies owing to significant differences in how earthquake hypocenters are reported before and after 2013. In reviewing the earthquake catalogs downloaded from the USGS Comcat database[36], we find that hypocenter depth errors are rarely reported before 2013. As a result, mean annual hypocenter depth prior to 2013 is calculated as a common arithmetic average with error bars corresponding to two standard errors of the mean. Beginning in 2013, the error associated with hypocenter depth is reported, thus

mean annual hypocenter depth ($z_{avg}$) between 2013 and 2018 is weighted by the inverse square of the reported depth error:

$$z_{avg} = \frac{\sum z_i/\sigma_i^2}{\sum 1/\sigma_i^2} \qquad (8)$$

where $z_i$ is reported hypocenter depth and $\sigma_i$ is the associated hypocenter depth error. The corresponding error on the mean ($\sigma^2(z_{avg})$) is given by: $\sigma^2(z_{avg}) = 1/(\sum 1/\sigma_i^2)$. To clearly differentiate between the arithmetic mean and error-weighted mean calculations, the arithmetic mean is presented in Fig. 1 as open circles (1995–2012), while the error-weighted mean is presented as solid black circles (2013–2018). It is important to note that mean annual hypocenter depths lacking error bars in the interval 1995–2012 arise because shallow earthquakes were commonly reported to occur at 5 km depth (without depth error) before 2013, and, as a consequence, these results should be considered less reliable.

**Earthquake frequency-magnitude analysis**. The 2013–2018 earthquake catalog for north-central Oklahoma and southern Kansas (36.125° to 37.235° and −97.800° to −99.602°) was initially subdivided by reported earthquake depth and separated into 1 km intervals. For each depth interval, a cumulative frequency-magnitude plot was generated and the Gutenberg-Richter $b$-value was determined as slope of the least squares regression fit to each distribution (Supplementary Fig. 8).

## Data availability

All data used for this study are freely available to the public from a variety of sources. USGS earthquake catalog: https://earthquake.usgs.gov/earthquakes/search/. USGS National Produced Water Database: https://energy.usgs.gov/Portals/0/Rooms/produced_waters/tabular/USGSPWDBv2.3c.csv. Oklahoma injection well data 2010 - present: http://www.occeweb.com/og/ogdatafiles2.htm. Oklahoma injection well data by county from 1995–2010 are available upon request from the Oklahoma Corporation Commission or RMP. New Mexico injection well volume per county: https://wwwapps.emnrd.state.nm.us/ocd/ocdpermitting/Reporting/Production/CountyProductionInjectionSummary.aspx. Colorado injection well data: https://cogcc.state.co.us/data.html#/cogis.

## Code availability

All wastewater injection models were completed with the TOUGH3 numerical simulation code, which is publicly available from Lawrence Berkeley National Laboratory at: https://tough.lbl.gov/software/tough3/. Supplementary Fig. 1 was produced with ParaView 5.5.0: https://www.paraview.org/download/. All other data were plotted with Generic Mapping Tools v5: http://gmt.soest.hawaii.edu.

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

## Acknowledgements

R.M.P. thanks Professor Emeritus J.D. Rimstidt for constructive comments on an early draft of this manuscript. This study is based upon work supported by the U.S. Geological Survey under Grant No. G19AP00011. The views and conclusions contained in this document are those of the authors and should not be interpreted as representing the opinions or policies of the U.S. Geological Survey. Mention of trade names or commercial products does not constitute their endorsement by the U.S. Geological Survey. Mr. Vicente Vasquez generously provided Oklahoma Corporation Commission underground injection data for the period 1995–2010. Computational resources were provided by Virginia Tech Advanced Research Computing.

## Author contributions

R.M.P. is responsible for study design, methods, analysis, and manuscript preparation. R.S.J. assisted with numerical model development. H.W. analyzed earthquake hypocenter data. M.C.C. provided technical oversight on earthquake data analysis.

## Additional information

**Competing interests:** The authors declare no competing interests.

