## [Peer Review File · Nature Communications]

Reviewers' comments:

Reviewer #1 (Remarks to the Author):

The concept of this manuscript is terrific. Truly novel, and worthy of high praise. It follows nicely on previous work, and the premise (induced earthquake events) is truly timely.

However, the approach -- a model simulation analysis with effectively homogeneous permeability, a synthetic density variability model, and an extremely coarse model grid with limited comparison to field data, seemed underwhelming. Indeed, I was excited by the premise as I read the initial few pages of the manuscript, but then realized that the conclusions are predicated nearly exclusively on the modeling analysis rather than on actual observations. I understand that actual observations are extremely limited, but a more fundamental analysis of a detailed small site with high resolution actual data and measured observations may be more compelling than a very coarse 3-D model scaling hundreds of kilometers. I understand that using effectively homogeneous permeability is essential to isolate the role of density differentials, but that begs the question whether a 1-D model would suffice.

If this paper could be effective WITHOUT the numerical modeling analysis, more focus on observed data and fundamental calculations to support the correlation between observations and the physics of flow, then I recommend re-writing it with a new analysis. The modeling (as it is, coarse with only limited calibration to actual data) reduces the impact of the message of the paper, or so is my opinion. Reliance on the models and simulated results to confirm conclusions drawn seems too speculative. Again, the concept as posed is novel and new, and worthy of praise!

Reviewer #2 (Remarks to the Author):

Pollyea et al. present a convincing investigation into the role of variable fluid density on salt water disposal induced earthquakes, a relevant topic that is of interest to a broad community. The authors demonstrate that the contrasting high density of injected wastewater versus basement fluids can increase subsurface fluid pressures after injection rate are reduced, potentially causing fewer, but higher magnitude, earthquakes. This work is both timely for the ongoing mitigation of induced seismicity and has scientific merit, with the potential to redefine and improve the factors consider in wastewater induced seismicity. I recommend prompt publication after a few minor issues, listed below, are addressed.

While the authors present a case in the Raton Basin where a lack of density differences fail to cause deeper earthquake epicenters, this paper would benefit from examining how large the density contrasts between the injected wastewater and basement fluids need to be to propagate deeper earthquake hypocenters. Additionally, this paper may relate to a larger, global audience by noting how many salt water disposal basins experience high-density contrasts worldwide (i.e., is this problem isolated to only Kansas and Oklahoma?).

L36-38, Sentence should be rewritten for clarity; it would be helpful to provide a range (or order of magnitude approximation) of how density and viscosity vary between wastewater and host rock fluids.

L77, What time step is used?

Move FS7 to main text as this is a major figure, move F4 to supplementary information or add contours from F4 to F2.

Reviewer #3 (Remarks to the Author):

This is a nice study, showing the effect of density-dependent flow due to oilfield wastewater disposal, on the occurrence of seismic events. The paper is well-written and clear and the methods are comprehensively described. The research is novel and leads to a better understanding of an important subject.

For the discussion, it might be interesting to consider whether dilution of the injected wastewater, to achieve the same fluid density prior to injection, would be a better strategy for reducing long-term seismicity.

The authors should also explain more accurately the effect of the decreasing b-value.

Line 174: "The implication of lower b-values at 8+ km depths is that the probability of high-magnitude earthquakes is larger than at shallower depths."

This is not necessarily true. Whilst the probability of any one event being of a higher magnitude increases with the decreasing b-value (i.e. increasing depth), the probability of occurrence of a higher magnitude event may not, since the total number of events per year is also decreasing. So for example, whilst in Figure 5c, there is an increase in the number of higher magnitude events from 2017 to 2018, in the two years preceding that (when the denser fluids are at shallower depths) the number of higher magnitude events is larger. So there is a trade-off between the increasing proportion of high magnitude events and the reduction in the total number of events.

Rebecca Lunn .

Reviewer #1 (Remarks to the Author):

The concept of this manuscript is terrific. Truly novel, and worthy of high praise. It follows nicely on previous work, and the premise (induced earthquake events) is truly timely.

However, the approach -- a model simulation analysis with effectively homogeneous permeability, a synthetic density variability model, and an extremely coarse model grid with limited comparison to field data, seemed underwhelming. Indeed, I was excited by the premise as I read the initial few pages of the manuscript, but then realized that the conclusions are predicated nearly exclusively on the modeling analysis rather than on actual observations. I understand that actual observations are extremely limited, but a more fundamental analysis of a detailed small site with high resolution actual data and measured observations may be more compelling than a very coarse 3-D model scaling hundreds of kilometers. I understand that using effectively homogeneous permeability is essential to isolate the role of density differentials, but that begs the question whether a 1-D model would suffice.

If this paper could be effective WITHOUT the numerical modeling analysis, more focus on observed data and fundamental calculations to support the correlation between observations and the physics of flow, then I recommend re-writing it with a new analysis. The modeling (as it is, coarse with only limited calibration to actual data) reduces the impact of the message of the paper, or so is my opinion. Reliance on the models and simulated results to confirm conclusions drawn seems too speculative. Again, the concept as posed is novel and new, and worthy of praise!

AUTHOR RESPONSE: We thank Reviewer #1 for the thorough review of our study and for the generous praise. The Reviewer raises some excellent points about our numerical simulation methods, which we address with our responses to specific comments (below). However, the broad theme of this review implies that multi-physics numerical simulation provides little value for testing the hypothesis that fluid density contrasts drive pressure transients into the seismogenic zone. We respectfully disagree with this sentiment and hope the following discussion provides a compelling (or perhaps acceptable) rationale for why numerical simulation is appropriate for this study.

Our study presents a mechanistic explanation for the observation that earthquake hypocenters are systematically deepening in Oklahoma, USA. It is important to recognize that *the observation itself is new*, and it is based on our analysis of the USGS Comprehensive Earthquake Catalog (USGS, 2019) using county-level data aggregation. We chose county-level data aggregation because unconventional oil and gas plays are characterized by rapid deployment of numerous, closely spaced production and injection wells with relatively short (3 – 5 yr) life spans, so their geographic footprint changes rapidly. As a result, the hypocenter signal becomes obscured when we analyze larger regions, but the signal also becomes excessively noisy when we tried to isolate on a single injection well. We also find that the deepening hypocenter trend is persistent within both of the major oil and gas plays in Oklahoma, but we do not see this trend in Raton Basin of New Mexico and Colorado. In order to explain this discrepancy, we considered the differences between the two regions and realized that wastewater produced in Oklahoma comes from the Mississippi Lime (known to be highly brackish), while wastewater from the Raton Basin comes from coalbeds. We then analyzed the USGS National Produced Waters Geochemical Database (Blondes et al., 2017) to learn about the fluid composition of each wastewater, and found that they are characterized by substantially different total dissolved solids (TDS) concentrations – wastewater from the Mississippi Lime in Oklahoma and Kansas has very high TDS concentration, while wastewater from coalbed deposits in the Raton Basin has very low TDS concentration. We continued analyzing the produced waters

database to find constraints on TDS concentration in the Precambrian basement (seismogenic zone) - samples collected from basement rocks exhibit much lower TDS concentration than produced waters in Oklahoma, but higher TDS concentration than produced waters in the Raton Basin. In combination, this **data analysis** led us to the hypothesis that high density wastewater can drive pressure transients into the seismogenic zone, thus explaining the systematically deepening hypocenter trend.

Based on the discussion above, we are confident in our hypothesis; however, we cannot make direct observations to support it. This is because earthquakes in Oklahoma, Kansas, and the Raton basin generally occur at depths between 4 and 10 km, where there are no current hydrogeologic data and to our **knowledge there are no wells completed to these depths in these regions**. Moreover, a recent analysis by Lukawski et al. (2014) suggests that wells completed from 4 to 10 km depth **cost \$10M - \$30M (9 - 27M €) each**. This is clearly cost prohibitive. Even if such wells were available (they aren't) or we could afford to install them (we can't) they would be of little value at present because the areas under investigation for this study have been subject to oilfield wastewater disposal for last 5 – 10 years. As a consequence, the fluid pressure (or hydraulic head) signal would be obscured by the presence of oilfield wastewater already present in the seismogenic zone.

As with many other research areas in the natural sciences (paleoclimate, planetary interiors, etc.), we chose to test our hypothesis with numerical simulation based on what we know about the physical and chemical processes involved, as well as what we can reasonably infer about hydraulic properties in the seismogenic zone (4 – 10 km depth for this study). In the context of oilfield wastewater disposal, there is a well-established history of implementing numerical models of pore fluid pressure to explain earthquake occurrence because direct measurements of the hydrogeologic system cannot be acquired within the seismogenic zone. Keranen et al. (2014) is the landmark study for implementing numerical simulation to explain the relationship between pressure propagation and earthquake occurrence in Oklahoma, and many others have shown that numerical simulation yields convincing results, e.g., see Brown et al., 2017; Dempsey & Riffault, 2019; Goebel et al., 2017; Hearn et al., 2018; Keranen et al., 2014; Langenbruch et al., 2018; Langenbruch & Zoback, 2016; Norbeck & Horne, 2018; Ogwari et al., 2018; Zhang et al., 2013.

For each case cited above, there is no model calibration in the traditional sense because there are no hydrogeologic data within the seismogenic zone from which to calibrate. As a consequence, there is certainly some amount of speculation in the conclusions; however, this is balanced by the fact that mass and energy conservation yield fundamental equations that we can solve to estimate how fluid and heat flow through geologic systems. Of course, the models are subject to parametric uncertainty, but the community broadly agrees that permeability is the first order control on pressure propagation during fluid injections. At the scales and depths of interest for this study, Townend and Zoback (2000) constrain bulk permeability of brittle crust to range from $\sim 10^{-16} - 10^{-17} \text{ m}^2$. Our study reproduces this range of crustal scale bulk permeability and we further invoke (i) the multiple interacting continua approximation (Pruess & Narasimhan, 1982) to account for hydraulic communication between interconnected fracture networks (fracture-fracture flow), as well as between fracture networks and solid matrix (fracture-matrix flow) and (ii) the crustal-scale depth dependent permeability model by Manning and Ingebritsen (1999). While this approach to representing the Precambrian basement is certainly subject to uncertainty, stochastic reservoir simulation methods cannot effectively reduce this uncertainty because these methods also require constraints, e.g., spatial correlation models, probability distributions, conditioning data, but none of these data exist for the seismogenic zone. Thus, stochastic reservoir simulation and Monte Carlo modeling combined with ensemble analytics add substantial computational overhead without a commensurate reduction in uncertainty.

Despite the parametric uncertainty in our simulations, we have carefully developed our models to reproduce as much as we know and can reasonably infer about the hydraulic properties of the seismogenic zone in northern Oklahoma. Moreover, our model reproduces the known geothermal heat flux in northern Oklahoma (Cranganu et al., 1998), and the well-established TOUGH3 simulation code (Jung et al., 2017) accounts for non-isothermal, variable density fluid flow, including PTX dependence of brine properties. This is the first ever numerical model to account for the complex fluid system properties that govern pressure transients during oilfield wastewater disposal, and we believe that our numerical simulations add a great deal of knowledge about the hydrogeology of injection-induced earthquakes. While we have little doubt that there is plenty of room to debate the simulation details, we strongly believe that our model identifies the mechanistic processes that explains our observations. And we do hope that the Reviewer will see the value in our application of numerical simulation to the problem of injection-induced earthquakes.

References

- Blondes, M.S., Gans, K.D., Engle, M.A., Kharaka, Y.K., Reidy, M.E., Saraswathula, V., Thordsen, J.J., Rowan, E.L., and Morrissey, E.A. 2017. US Geological Survey National Produced Waters Geochemical Database v2.3 Documentation. Internet download 10 March 2018 at: https://energy.usgs.gov/Portals/0/Rooms/produced_waters/tabular/USGSPWDBv2.3c.csv
- Brown, M.R., Ge, S., Sheehan, A.F. and Nakai, J.S., 2017. Evaluating the effectiveness of induced seismicity mitigation: Numerical modeling of wastewater injection near Greeley, Colorado. *Journal of Geophysical Research: Solid Earth*.
- Dempsey, D. and Riffault, J., 2019. Response of Induced Seismicity to Injection Rate Reduction: Models of Delay, Decay, Quiescence, Recovery, and Oklahoma. *Water Resources Research*, 55(1), pp.656-681.
- Goebel, T.H.W., Weingarten, M., Chen, X., Haffener, J. and Brodsky, E.E., 2017. The 2016 Mw5. 1 Fairview, Oklahoma earthquakes: Evidence for long-range poroelastic triggering at >40 km from fluid disposal wells. *Earth and Planetary Science Letters*, 472, pp.50-61.
- Hearn, E.H., Koltermann, C. and Rubinstein, J.R., 2018. Numerical models of pore pressure and stress changes along basement faults due to wastewater injection: Applications to the 2014 Milan, Kansas earthquake. *Geochemistry, Geophysics, Geosystems*.
- Jung, Y., Pau, G.S.H., Finsterle, S. and Pollyea, R.M., 2017. TOUGH3: A new efficient version of the TOUGH suite of multiphase flow and transport simulators. *Computers & Geosciences*, 108, pp.2-7.
- Keranen, K.M., Weingarten, M., Abers, G.A., Bekins, B.A. and Ge, S., 2014. Sharp increase in central Oklahoma seismicity since 2008 induced by massive wastewater injection. *Science*, 345(6195), pp.448-451.
- Langenbruch, C., Weingarten, M. and Zoback, M.D., 2018. Physics-based forecasting of man-made earthquake hazards in Oklahoma and Kansas. *Nature Communications*, 9(1), p.3946.
- Langenbruch, C. and Zoback, M.D., 2016. How will induced seismicity in Oklahoma respond to decreased saltwater injection rates? *Science Advances*, 2(11), p.e1601542.
- Manning, C.E. and Ingebritsen, S.E., 1999. Permeability of the continental crust: Implications of geothermal data and metamorphic systems. *Reviews of Geophysics*, 37(1), pp.127-150.
- Norbeck, J.H. and Horne, R.N., 2018. Maximum magnitude of injection-induced earthquakes: A criterion to assess the influence of pressure migration along faults. *Tectonophysics*, 733, pp.108-118.
- Ogwari, P.O., DeShon, H.R. and Hornbach, M.J., 2018. The Dallas-Fort Worth airport earthquake sequence: Seismicity beyond injection period. *Journal of Geophysical Research: Solid Earth*, 123(1), pp.553-563.
- Pruess, K. and Narasimhan, T.N., 1982. *Practical method for modeling fluid and heat flow in fractured porous media* (No. LBL-13487; CONF-820242-1). Lawrence Berkeley Lab., CA (USA).
- Townend, J. and Zoback, M.D., 2000. How faulting keeps the crust strong. *Geology*, 28(5), pp.399-402.
- U.S. Geological Survey (USGS), 2018. ANSS Comprehensive Earthquake Catalog (ComCat): <https://earthquake.usgs.gov/earthquakes/search/> accessed 21 February 2019.

Specific Comments

Author note: Comments from the annotated pdf document are reproduced here with their corresponding label.

CU1: No diffusion models suggest that pressure fronts cease advancing when the original pulse ceases. On the contrary, diffusion continues as long as a pressure differential exists. Analytical as well as numerical solutions to the diffusion equation yield such tails of decay.

AUTHOR RESPONSE: The reviewer is correct. Upon rereading the passage in light of this comment, it is clear that we overstated the pressure response when injections cease. The subject passage has been rewritten as:

This trend of increasing hypocenter depth years after substantial SWD volume reductions is unexpected because pressure diffusion models show that the rate of pressure accumulation decreases rapidly when injection operations cease \cite{langenbruch2010}.

CU2: What is the impact of using county-wide averages? Specifically, would formation- specific values provide different results? Why not simply use a sensitivity analysis, and ignore the county-averaged.

AUTHOR RESPONSE: The trend in which earthquake hypocenter depths systematically increase in Oklahoma is based on county-level data aggregation of the USGS Comprehensive Earthquake Catalog. The trend is not apparent in larger areas (e.g. state wide) due to the timing of unconventional oil & gas operations – production in Oklahoma started east of the Nemaha Fault Zone in the Hunton play around 2008-09, but the Anadarko Shelf in northern Oklahoma didn't see large-scale production until 2013-14. We also looked at finer-scale data aggregation – the trend is apparent, but much noisier than county-level. Since earthquakes in Oklahoma are induced by oilfield wastewater disposal, we settled on county-level averages for TDS concentration data because wastewater is reinjected close to its source. In addition, all TDS data for the Anadarko Shelf are from the **Mississippi Lime formation**, which is the oil/gas producing formation in the region. For the Hunton play (Oklahoma & Lincoln Counties), all TDS data are from the **Hunton formation**. *In other words, we sorted the TDS data by county and formation* – this is clearly presented in Table 1 and the “Data Sources” section of our Methods.

CU3: This study is purely modeling?

AUTHOR RESPONSE: No. Our hypothesis that the density differential between oilfield wastewater and fluids in the seismogenic zone is based on detailed analysis of the USGS Comprehensive Earthquake Catalog and the USGS National Produced Waters Geochemical Database. Our hypothesis test compares simulation results with earthquake hypocenter data. In other words, our simulations provide a mechanistic explanation for the observation that earthquakes are getting deeper in Oklahoma – the observation is based on data and the mechanistic process is based on our numerical model. We further analyze the earthquake record

in northern Oklahoma and southern Kansas to find that Gutenberg-Richter b -value decreases with depth.

CU4: Is this a hypothetical model scenario, or is this the actual depth of screening? If the latter, what is the actual well name and location? If the former, on what basis is the hypothetical scenario?

AUTHOR RESPONSE: The model is hypothetical, but it is based on numerous characteristics of oilfield wastewater wells and geological characteristics in Alfalfa County, Oklahoma. For example, we reproduce (i) the depth and thickness of the Arbuckle formation, (ii) known permeability and porosity of the Arbuckle formation, (iii) typical wastewater TDS concentration, (iv) typical wastewater injection rate, (v) typical well completion depth, (vi) regional geothermal heat flux in northern Oklahoma, and (vii) temperature- and composition-dependent properties of injection and host rock fluids. What we don't know about the hydrogeologic properties of the seismogenic zone in Oklahoma, we find reasonable approximations in the broader literature. For example, the permeability range of the seismically active crust is constrained by Townend and Zoback (2000) to range from $\sim 10^{-16} - 10^{-17} \text{ m}^2$, which we use to pin the lower end of effective permeability for our three model scenarios. We also know from unpublished presentations and discussions with researchers in Oklahoma that the uppermost basement rock is pervasively fractured, which is why we modeled the Precambrian basement as multiple interacting continua with fracture and matrix properties, as wells as depth-decaying fracture permeability. To provide additional context for this approach, the figure below shows the distribution of oilfield wastewater disposal wells in Oklahoma in 2015 with a callout specifically illustrating the well locations that are completed in the Arbuckle formation in Alfalfa County. ***Our model reasonably approximates any one of the red wells shown in the callout.***

CU5: Why this value?

AUTHOR RESPONSE: Typical value for unfractured crystalline rock (Freeze & Cherry, 1979, Table 2.2).

CU6: Why this value?

AUTHOR RESPONSE: There are no data to constrain the volume that fractures occupy in the seismogenic zone. However, we selected 2% volume for the fracture continuum with 10% porosity, which yields a quite reasonable estimate for effective fracture porosity of 0.2%. This is congruent with deep well testing by Stober and Bucher (2000), who found 0.5% effective porosity of crystalline basement fractures from the Urach 3 geothermal borehole in Germany.

Reference

Stober, I. and Bucher, K., 2000. Hydraulic Properties of the upper Continental Crust: data from the Urach 3 geothermal well. In *Hydrogeology of Crystalline Rocks* (pp. 53-78). Springer, Dordrecht.

CU7: Why these values?

AUTHOR RESPONSE: In our model, the basement fracture permeability decays with depth in accordance with the Manning and Ingebritsen (1999) model: $k(z) = k_0(z/z_0)^{-3.2}$, where k_0 is the permeability at depth z_0 . In our model, z_0 is 2,300 m depth, which corresponds with the Arbuckle-basement contact where permeability (k_0) is $5 \times 10^{-13} \text{ m}^2$. We also consider two additional basement permeability scenarios for which k_0 is $1 \times 10^{-13} \text{ m}^2$ and $5 \times 10^{-14} \text{ m}^2$. ***This is discussed in the Methods section of the manuscript and illustrated graphically in Figure S1b.***

CU8: Any use or application of actual permeability values? Or stochastic or other forms of heterogeneity?

AUTHOR RESPONSE: There are no *in situ* permeability data within the seismogenic zone of northern Oklahoma. However, Townend and Zoback (2000) constrain bulk permeability of brittle crust to range from $\sim 10^{-16} - 10^{-17} \text{ m}^2$, the result of which maintains hydrostatic conditions throughout much of the seismically active crust. We used this range to pin the lower end of effective permeability for our model scenarios.

We have **added the following passage** with the Townend and Zoback (2000) reference to the manuscript:

This basement permeability distribution is congruent with estimates for the seismically active crust suggesting that bulk permeability is on the order of $\sim 10^{-16}$ - 10^{-17} m^2 (Townend 2000).

We did consider stochastically generated permeability distributions for this study; however, sequential simulation methods require spatial correlation models, probability distributions, and conditioning data. This information does not exist for permeability distributions within the seismogenic zone of northern Oklahoma (or anywhere else). Although we could do fully unconditional and randomly generated permeability distributions, scale them by depth, and produce simulation ensembles for each model scenario in the study, the ensemble mean (e-type estimates) for this approach would converge on the three solutions we present in this study (assuming that the mean for each scenario is defined as given in Figure S1). Moreover, the ensemble variance would be a function of the probability distribution used to generate the random permeability distributions, and because this probability distribution is unknown the resulting uncertainty estimate would be of little practical value. In our opinion, the application of stochastic simulation methods requires tremendous computational expense without a commensurate decrease in model uncertainty.

CU9: Again, this is a hypothetical scenario, NOT based on actual data? If so, are any stochastic considerations invoked?

AUTHOR RESPONSE: We addressed this issue in our responses to comments CU4 and CU8.

CU10: A regional gradient, not local? Are local temperature anomalies a potential source of greater density contrasts?

AUTHOR RESPONSE: Yes, local temperature anomalies will affect the density contrast; however, these effects are vanishingly small in comparison to the density contrast that results from differences in the TDS load between wastewater and host rock fluids. Moreover, there are no data to constrain the location and magnitude of local thermal anomalies, if they exist in this system.

CU11: Why these particular values?

AUTHOR RESPONSE: We chose a 10-year injection period because injection-induced earthquakes have been common throughout Oklahoma since 2009, i.e. ten years ago. The 40-year recovery interval was chosen because it fully encompasses the time period for pressure accumulation and recovery in the primary model scenario.

CU12: How about the uncertainty in the reservoir unit?

AUTHOR RESPONSE: Permeability of the Arbuckle formation is relatively well constrained to be on the order of 10^{-13} m² (Kroll et al., 2017; Morgan & Murray, 2015). We have added these references when we present Arbuckle permeability.

Kroll, K.A., Cochran, E.S. and Murray, K.E., 2017. Poroelastic properties of the Arbuckle Group in Oklahoma derived from well fluid level response to the 3 September 2016 M w 5.8 Pawnee and 7 November 2016 M w 5.0 Cushing earthquakes. *Seismological Research Letters*, 88(4), pp.963-970.

Morgan, B. C., and K. E. Murray, 2015. Characterizing small-scale permeability of the Arbuckle Group, Oklahoma, Okla. Geol. Surv. Open-File Rept. OF2-2015, 1–12.

CU13: [No comment given]

CU14: You simulated a regional, uniform value of temperature gradient, or local values based on well data?

AUTHOR RESPONSE: There is no well data for the seismogenic zone and temperature data for injection wells is obscured by many years of wastewater disposal. We modeled the geothermal gradient on the basis of the known heat flux, which is 40 mW m⁻² (Cranganu et al., 1998). **This is discussed in the Methods section.**

CU15: Why 2 years? Would different density values affect your simulated time lag results? Or the diffusivity values resulting from effectively homogeneous formations?

AUTHOR RESPONSE:

Q1: Two years is what came from the simulation results, we did not choose this result. However, this result does vary by permeability distribution, which we discuss in the manuscript.

Q2: Different density values clearly affect this result – when there is no density contrast, there is no upward change in the ΔP_f vs time curve at 4, 5, and 6 km depths. Figure 3 demonstrates this very clearly, i.e., compare the solid and dashed lines, the latter of which are the constant density model.

Q3: First, hydraulic diffusivity (D) is temperature dependent owing to the viscosity term in hydraulic conductivity. As such, we don't think hydraulic diffusivity is an effective parameter for describing

the hydraulic properties of the seismogenic zone. Second, our permeability distribution is not homogeneous or even “effectively homogeneous” because (i) each grid cell comprises both fracture and matrix permeability, which are different, so our grid cells themselves are heterogeneous and (ii) effective permeability in the basement decreases with depth so it is not the same everywhere, which means the basement permeability distribution is heterogeneous. And finally, ΔP_f vs time is clearly dependent on the permeability distribution. We present this in lines 124 – 127 of the revised manuscript with supporting results in the supplemental.

CU16: A regional gradient, not local? Are local temperature anomalies a potential source of greater density contrasts?

AUTHOR RESPONSE: We discuss this in our response to comment CU14.

CU17: Demonstrated only by modeling, or also by measured observations?

AUTHOR RESPONSE: There are no pore fluid pressure measurements in the seismogenic zone (4+ km depth). This result comes from our models.

CU18: “hazard” or “hazards”?

AUTHOR RESPONSE: hazard

CU19: Beautiful. Love data.

AUTHOR RESPONSE: We agree, and we analyze *a lot* of data for this study.

CU20: Modeling is also effective, but most effective when parameterized and predicated on high resolution data, including measured observations.

AUTHOR RESPONSE: We agree that modeling is effective. For well constrained sites, models can be used as predictive tools. For poorly constrained sites, models can be used to identify the mechanistic processes that result in observable phenomena. Our study uses this latter approach. Specifically, we **observe** that earthquake hypocenters systematically deepen when oilfield wastewater comprises higher density than basement fluids, and our **models demonstrate** that density-driven pressure transients migrate at comparable rates to the earthquake trend. We think this is an effective use of multi-physics numerical simulation.

CU21: So, a low-density contrast model? Would a well-designed sensitivity analysis accomplish a well-informed set of conclusions?

AUTHOR RESPONSE: Yes, we are comparing the behavior of a constant density model with an identically parameterized model comprising variable density fluids. Please see lines 78 – 82, Figures 2 – 4 of the revised manuscript and Figures S2 – S7 of the supplement.

CU22: This is very coarse resolution, at first glance. But, such is consistent with homogeneous permeability (effectively) and a synthetic density distribution. Even with local grid refinement, the cell size must be at least 100 m at best, close to the wells, with 3 km or more in size at far distances from the well. How demonstrative is such a coarse resolution model? Would not a 1-D model suffice?

AUTHOR RESPONSE:

Q1: Lateral grid discretization near the well is 30 m and increases to ~100 m within 15 km of the injection well. In the far field, lateral grid discretization coarsens substantially; however, our study focuses on pressure transients near the well at depths between 4 and 6 km. Moreover, the 10 kPa ΔP_f contour does not extend beyond 20 km, so the coarse far-field grid discretization does not affect our results or conclusions.

Q2: No, a 1-D well would not suffice because we are interested both vertical and lateral pressure propagation caused by density-driven pressure transients in the seismogenic zone – we discuss lateral pressure propagation in lines 98 – 100 of the manuscript. Moreover, it is not possible to simulate real-world injection volume in a 1-D model. That said, we could have used a 2-D radially symmetric model domain with cylindrical grid cells, the latter of which allow for physically realistic injection volume. But we decided on the 3-D unstructured grid with 4-fold symmetry because we can use this grid to study the effects of multiple injection wells operating in close spatial proximity, as is the case in northern Oklahoma (see image presented in response to CU4 – this is a future research direction for our team).

CU23: Model description (all that's highlighted) can be cited from the TOUGH3 User's manual; it's superfluous here.

AUTHOR RESPONSE: We agree, but our experience is that readers appreciate a summary of the governing equations that are being solved. We'll have this discussion with the editorial team, if the paper is accepted for publication.

CU24: These are averages for each (entire) county?

AUTHOR RESPONSE: Yes, this is stated in the title of *Table 1: Composition of water produced from Mississippi Lime, Hunton, and Precambrian (basement) formations in select counties of Oklahoma and Kansas.*

Reviewer #2 (Remarks to the Author):

Pollyea et al. present a convincing investigation into the role of variable fluid density on salt water disposal induced earthquakes, a relevant topic that is of interest to a broad community. The authors demonstrate that the contrasting high density of injected wastewater versus basement fluids can increase subsurface fluid pressures after injection rate are reduced, potentially causing fewer, but higher magnitude, earthquakes. This work is both timely for the ongoing mitigation of induced seismicity and has scientific merit, with the potential to redefine and improve the factors consider in wastewater induced seismicity. I recommend prompt publication after a few minor issues, listed below, are addressed.

AUTHOR RESPONSE: We thank Reviewer #2 for the thorough review of our study and for the encouraging remarks.

While the authors present a case in the Raton Basin where a lack of density differences fail to cause deeper earthquake epicenters, this paper would benefit from examining how large the density contrasts between the injected wastewater and basement fluids need to be to propagate deeper earthquake hypocenters.

AUTHOR RESPONSE: We agree with the reviewer that this study leaves open a number of important questions about the parametric controls on density-driven pressure transients. However, the present study is designed to (i) introduce the hypothesis and observations that density-driven pressure transients drive earthquakes systematically deeper, (ii) provide a mechanistic explanation for the observed phenomenon, and (iii) explore the consequences of deeper earthquakes. We think that a broader study on the parametric controls of density-driven pressure transients is beyond the scope of this study; however, we are in the early stages of a comprehensive study along these lines.

Additionally, this paper may relate to a larger, global audience by noting how many salt water disposal basins experience high-density contrasts worldwide (i.e., is this problem isolated to only Kansas and Oklahoma?).

AUTHOR RESPONSE: At this point, we only have evidence to support density-driven pressure transients in Oklahoma and Kansas, presumably because nowhere else in the world has experienced oilfield wastewater disposal on such an enormous scale. In the context of reaching a larger global audience, there are certainly intra-plate basins throughout the world that likely have high density fluids and there are likely data available to support this claim (although we don't have data anywhere except the United States); however, we are much less confident that there are TDS concentration data in the underlying basement rocks. For example, the USGS National Produced Waters Database has 114,944 records, but contains only 18 records with TDS concentration within Precambrian basement rocks – fortunately these records are near the Anadarko Shelf and Raton Basin (which we use in this study). In our opinion, taking on a global analysis of density contrasts between basin brine and basement fluids is likely to be a time-consuming endeavor with a low probability of success, so we respectfully decline this suggestion. Nevertheless, we do agree that there is a broad global audience for this study, so we added the following closing statement to our discussion:

In closing, we note that recent global estimates of unconventional fossil fuel resources exceed 440 billion tons oil and 227 trillion cubic meters gas from 363 petroleum basins worldwide

(Hongjun et al., 2016). As these resources are developed, results from this study suggest that the density contrast between produced waters and basement fluids is a fundamental component of the risk profile for injection-induced earthquakes during oilfield wastewater disposal in deep geologic formations. We hope this study motivates the further research into the relationship between fluid properties and injection-induced seismicity.

L36-38, Sentence should be rewritten for clarity; it would be helpful to provide a range (or order of magnitude approximation) of how density and viscosity vary between wastewater and host rock fluids.

AUTHOR RESPONSE: We clarified (and added details to) the subject passage as follows:

A common attribute among these and other modeling studies \cite{brown2018, norbeck2018b, zhang2013} is that fluid properties (e.g., density and viscosity) are assumed to be identical between the wastewater and host rock fluids. However, SWD operations drive pressure transients over km scales into the seismogenic zone, where fluid properties vary substantially due to thermal and geochemical variability. For example, at pressure and temperature conditions representative of ~5 km depth (50 MPa and 25°C) the density of pure water is 1,018 kg m⁻³, but for brine composition of 200,000 parts per million NaCl the fluid density is 1,165 kg m⁻³ \cite{lvov1990} and the viscosity increases ~50%.

L77, What time step is used?

AUTHOR RESPONSE: We use automatic timestep control. If the simulation achieves convergence within four Newton-Raphson iterations, then the next time step doubles; however, if convergence is not achieved within eight Newton-Raphson iterations, then the current timestep is halved. **We added this detail to the Methods section.**

Move FS7 to main text as this is a major figure, move F4 to supplementary information or add contours from F4 to F2.

AUTHOR RESPONSE: We thank the reviewer for this suggestion. **We combined Figure S7 with Figure 1 so that all earthquake hypocenter depth trends are now on Figure 1. We also moved Figure 4d-f to the supplement as the new Figure S2** because the recovery phase of the constant density model adds little value to the main text, but we wanted to retain the information for the sake of completeness.

Reviewer #3 (Remarks to the Author):

This is a nice study, showing the effect of density-dependent flow due to oilfield wastewater disposal, on the occurrence of seismic events. The paper is well-written and clear and the methods are comprehensively described. The research is novel and leads to a better understanding of an important subject.

AUTHOR RESPONSE: We extend sincere gratitude to Dr. Lunn for the encouraging remarks!

For the discussion, it might be interesting to consider whether dilution of the injected wastewater, to achieve the same fluid density prior to injection, would be a better strategy for reducing long-term seismicity.

AUTHOR RESPONSE: This is an interesting idea, but we're not sure it's practical. The most effective way to dilute the high-density wastewater would be the addition of pure water, which would have to come from shallow aquifers. And given the enormous volume of oilfield wastewater that is being reinjected in Oklahoma and Kansas, the water demand would most certainly cause an adverse effect on the already stressed aquifers (one of which is the Ogallala Aquifer). Another option would be to use lower TDS basement fluids for dilution, but this would require very deep wells to access and 4:1 or greater mixing ratios – the economic and energy constraints seem highly unfavorable for this approach.

The authors should also explain more accurately the effect of the decreasing b -value.

Line 174: "The implication of lower b -values at 8+ km depths is that the probability of high-magnitude earthquakes is larger than at shallower depths."

This is not necessarily true. Whilst the probability of any one event being of a higher magnitude increases with the decreasing b -value (i.e. increasing depth), the probability of occurrence of a higher magnitude event may not, since the total number of events per year is also decreasing. So for example, whilst in Figure 5c, there is an increase in the number of higher magnitude events from 2017 to 2018, in the two years preceding that (when the denser fluids are at shallower depths) the number of higher magnitude events is larger. So there is a trade-off between the increasing proportion of high magnitude events and the reduction in the total number of events.

AUTHOR RESPONSE: We sincerely thank Dr. Lunn for catching this error. She is correct that the probability of a given earthquake magnitude is dependent on both the b -value and earthquake rate. We have revised this sentence to be technically correct:

The implication of lower b -values at 8+ km depths is that the proportion of high magnitude earthquakes increases relative to the earthquake distribution at shallow depths.

To make the connection between density-driven pressure transients, depth-decreasing b -values, and earthquake hazard, we added the following text to close the section:

Moreover, the persistence of density-driven fluid pressure transients may help to explain why the USGS one-year seismic hazard forecast found that the probability for damaging ground motion in Oklahoma and Kansas increased from 2017 to 2018 despite the broad trend of decreasing earthquake rates that began in 2016³³.

REVIEWERS' COMMENTS:

Reviewer #1 (Remarks to the Author):

Note to authors: I was looking forward to a more-detailed data analysis, rather than defense of numerical model application. Models can always be adapted to produce a desired result; data speak effectively and unequivocally. And, the models developed for this analysis utilize effectively homogeneous permeability, a synthetic density variability model, and an extremely coarse model grid. I stand by my original assessment that the results are underwhelming and the model simulation results are not compelling.

Reviewer #2 (Remarks to the Author):

I am content with the revisions of the manuscript since the previous version and recommend that the manuscript is published in its present form.

Reviewer #1 (Remarks to the Author):

Note to authors: I was looking forward to a more-detailed data analysis, rather than defense of numerical model application. Models can always be adapted to produce a desired result; data speak effectively and unequivocally. And, the models developed for this analysis utilize effectively homogeneous permeability, a synthetic density variability model, and an extremely coarse model grid. I stand by my original assessment that the results are underwhelming and the model simulation results are not compelling.

AUTHOR RESPONSE: We thank Reviewer #1 for again reviewing our study. Nevertheless, we respectfully disagree with the Reviewer's assessment of our study. As there are no remaining suggestions for improvement, we are making no further revisions and we leave the decision with the editor.

Reviewer #2 (Remarks to the Author):

I am content with the revisions of the manuscript since the previous version and recommend that the manuscript is published in its present form.

AUTHOR RESPONSE: We thank Reviewer #2 for the thorough review of our study and for the encouraging remarks.

Reviewer #3: No comments provided